# A service provider perspective in Irish horseracing on the availability of retirement specific support services for professional jockeys and perceived barriers and facilitators to their use

Laura Langton [1]*, Sarah Jane Cullen[2], Giles Warrington [3,4], Jennifer Pugh[5], Adrian McGoldrick[5], Ciara Losty[1]

**1** Department of Sport and Exercise Sciences, South East Technological University, Waterford, Ireland, **2** School of Health and Human Performance, Dublin City University, Dublin, Ireland, **3** Physical Education and Sport Sciences, University of Limerick, Limerick, Ireland, **4** Sports and Human Performance Research Centre, Health Research Institute, University of Limerick, Limerick, Ireland, **5** Irish Horseracing Regulatory Board, Kildare, Ireland

* laura.langton@postgrad.wit.ie

## Abstract

Retirement can come at any time in horse racing due to the unpredictable nature of the sport. Preplanning for the retirement transition in sport has been reported to lead to faster adjustment to retirement and better transition for athletes. The purpose of this study was to highlight the current support services available to Irish horseracing jockeys to aid with career transitions and to a) explore perceived barriers and b) perceived facilitators to their use. Semi-structured interviews were conducted with 11 representatives from various racing bodies including the Irish Horse Racing Regulatory Board (IHRB), Horse Racing Ireland (HRI), Racing Academy Centre of Education (RACE), Irish Jockeys Trust (IJT), The Jockeys Association (JA), Injured Jockeys Fund (IJF), Equipp and The Jockeys Pathway. Questions included, demographic information, perceptions of current use of support services, perceptions of barriers preventing jockeys from using support services and perceived facilitators to the use of support services in the future. Reflexive thematic analysis was used to analyse qualitative data that included inductive and deductive approaches. The higher order themes identified as barriers to use of transition support services, were the culture within the Irish horseracing industry, and insufficient cross organisation communication. Higher order themes relating to facilitators of engagement were accessible further education opportunities for jockeys, normalising career planning and the development of an exemplar support service for jockey career transitions. Normalising career planning and implementing an exemplar support programme for retiring jockeys is vital to ensure jockey wellbeing needs are met as they transition at the end of their racing career.

which permits unrestricted use, distribution, and reproduction in any medium, provided the original author and source are credited.

**Data availability statement:** Data for this study contain potentially identifying informations. Due to the small size of the Irish horseracing industry it was deemed necessary by the SETU ethics committee that data be accessed directly from them on request if needed (HealthSciencesResearchEthics.WD@setu.ie).

**Funding:** The author(s) received no specific funding for this work.

**Competing interests:** The authors have declared that no competing interests exist.

## Introduction

Elite athletes will go through many transitions within their sporting career [1–8]. These may include underage to adult level, junior to senior level, amateur to elite level and national to international level. Not all athletes will make each of these transitions in their sporting career. However the one transition common amongst all elite athletes is the eventual retirement transition. For some athletes this decision will be a conscious timely one that is planned by the athlete themselves' and by their support team, however this unfortunately is not always the case [9]. For many athletes, retirement can come earlier than they would like, through career ending injury and de-selection [10–11]. This is especially true in jockeys due to the high risk nature of the sport of horse racing which includes a high prevalence of falls and injuries [12]. Career demands of a jockey are high, with long working hours [13] and high levels of commitment [14] expected. It is, therefore of vital importance jockeys are supported through the retirement transition and that pre-planning is in place.

No research has been conducted in the horse racing industry in Ireland to assess the use of current support services. The main objectives of this research is to firstly identify if Irish jockeys are using the current support services that are available to them, secondly to identify perceived barriers to the use of support services and thirdly to identify perceived facilitators to use and improvements to the support services that could be beneficial.

To the authors knowledge only one study on retirement in jockeys have been previously published [9]. An Australian based industry report [9] focused on the welfare of 72 retired jockeys. Ninety percent of the jockeys believed that the development of a programme that offers social opportunities for retired jockeys would be beneficial for those who leave the industry. Thirty percent of the jockeys felt that they had received little or no support from the racing industry once they retired. Some of the issues associated with retirement in the jockeys included difficulties forming an identity away from the racing industry (28–38%), emotional distress (19%), loss of confidence (10%), and for a small number of jockeys, alcohol (5%) and gambling (2%) issues. Results from this Australian study highlight the problems that can arise post-retirement if support services are not in place to aid transitioning jockeys [9]. Preplanning for the retirement transition and partaking in dual careers could help prevent the problems that arise post-retirement for athletes.

Planning for retirement and partaking in dual careers has been reported as a crucial factor to enable a smooth and faster adjustment to retirement in elite athletes [15–16]. Adjustment to retirement has been found to be less successful when unplanned and unexpected [17–18]. Career development planning [16,19,20] and practical resource training [21] during the athlete's career have been found to aid in their transition out of competitive sport and to make it a more positive experience. Research has found multiple psychological benefits of dual careers for elite athletes including, a more balanced lifestyle, broader identity development, self-esteem, and better preparedness for athletic retirement. The pursuit of a dual career could also alleviate the high levels of athletic identity [22] that have been found to have a negative effect on career planning. Social benefits have also been found including,

expanded social networks and a larger support system [23]. Finally financial benefits have been found including, broader skills and a better chance of employment [24,25].

It has been found that athletes themselves tend not to plan for retirement unless it is provided by support systems from national governing bodies (NGB's) of their sport [26]. Athletes who are supported effectively by partaking in psycho-educational interventions focused on diversifying athletic identity, initiating the grief process, developing coping skills, identifying supports and reviewing mental health resources prior to their retirement transition are more resilient to the retirement transition [19–21,25,27–31]. It is evident that NGB's across sporting codes should be implementing pro-grammes to support athletes in career transition, in alignment with the International Society of Sport Psychology Position Stand, which acknowledges that athletes should be facilitated by their NGB's to pursue career planning and dual careers [32]. This service is currently in development within Irish horse racing as part of the Jockey Pathway, it is therefore hoped that this research will help guide the implementation and content of services provided in the future.

This research was guided by Schlossberg's Transition Theory Schlossberg [33] conceptualising transitions as events that require individuals to adapt to changes in roles, routines, relationships, and identities. Central to this model is the "4S system" (situation, self, support, and strategies), which proposes that an individual's capacity to cope with transi-tion is formed by personal characteristics, support system availability, and coping resources. Within professional sport, retirement symbolises a significant career and identity transition that may require structured support services to facilitate successful adjustment to post sporting life. Using this framework as a theoretical guide, the present study explores how existing support services within Irish horseracing contribute to the "support" dimension of the transition process, while also identifying perceived barriers and facilitators influencing jockeys' engagement with these services as they prepare for retirement.

The aim of this study was to explore the perceptions of key stakeholders involved in the delivery of support services within Irish horseracing regarding the current provision of career and retirement support for jockeys. Specifically, the study sought to address the following research questions:

RQ1: What support services are currently available to assist Irish jockeys in preparing for retirement from professional horse racing?

RQ2: What barriers do stakeholders perceive as limiting Irish jockeys' engagement with these support services?

RQ3: What facilitators do stakeholders perceive as promoting Irish jockeys' engagement with support services in prepa-ration for retirement?

Further research can then explore the thoughts of both current and retired jockeys on this topic, to gain an even deeper understanding and alternative perspective. Practical implications and evidence-based recommendations can then be pro-vided to the regulatory bodies within Irish horse racing to aid in the development of a bespoke career transition service to alleviate the stressors reported in previous research on Irish jockeys [13].

## Methodology

### Study design

This study was underpinned by ontological relativism and an emic epistemology. The experiences and meanings attributed by industry support service providers were interpreted as culturally situated within the Irish horse racing envi-ronment, rather than evaluated against external theoretical frameworks. Guided by Braun and Clarke's Big Q paradigm [34], data was viewed as co-constructed through interaction between researcher and participant, with both positioned as "insiders" due to their shared experience of working with jockeys. The lead author's prior engagement with the Irish jockey cohort enabled a reflexive use of subjectivity, enhancing interpretive depth, while maintaining methodological rigour. Through semi-structured interviews, the study sought to generate an in-depth, contextually grounded understanding of perceived barriers and facilitators to support service utilisation, recognising that multiple, experience-dependent realities exist [35].

## Participants

To give context to this research it is important to highlight that the Irish horse racing industry is compiled of two main governing bodies, namely the Irish Horseracing Regulatory Board (IHRB) and Horse Racing Ireland (HRI). These two bodies provide guidance on rules and regulations, jockey licensing and they provide support services to jockeys. IHRB and HRI also work closely with the other stakeholders and charities within the Irish horseracing industry (Jockeys Association, Injured Jockeys Fund, Racing Academy and Centre of Education and Jockeys Trust). As part of HRI there are two sub divisions; Equipp and the Jockey Pathway. Equipp offer grant programmes for further education, curriculum vitae preparations courses, interview preparation courses, a careers map detailing within industry career paths. The Jockey Pathway provides a sports psychology service and a careers coach. The Jockeys Association provide a pension scheme for jockeys, Injured Jockeys Fund provide funding for further education, Racing Academy and Centre of Education provide skills courses, such as thoroughbred administration, communications and horse breeding courses. Finally, the Jockeys Trust provide a career coach, a counsellor and funding for pursuit of further education when injured.

Eleven industry stakeholders in Irish horse racing participated in the present study (4 female, 7 male). Interviews took place from 11/11/2021− 14/12/2021. All key stakeholders (Irish Horse Racing Regulatory Board (IHRB), Horse Racing Ireland (HRI), the Jockey Pathway, RACE, Irish Injured Jockeys (IIJ), the Irish Jockeys Trust (IJT) and Equipp) within the Irish horse racing industry were represented. The key criterion for participant inclusion in this study was the need to have been working in their current role for at least one year. This criteria was applied so that an in-depth knowledge of supports service availability within their organisation and across organisations was optimised. Participants included those who actively provide support services to jockeys, those who organise support service provisions, as well as those in management roles within their organisations.

## Procedure

Ethical approval was granted from the South East Technological University (SETU) ethics committee (Ref: WIT2021REC029). All stakeholders listed above were contacted by email and were asked to forward the study information sheet to a representative in their organisation knowledgeable on support service provision to jockeys in preparation for retirement. Participants, if interested then volunteered to partake by emailing the researcher directly.

Prior to commencement of interviews, an information sheet and an informed consent form was sent to each participant via email. These were signed and returned to the researcher by email. A semi-structured interview guide was developed in line with the framework recommended by [36]. Semi-structured interviews were used for this study to ensure that open ended questions asked could be explored in detail and so that clarification and further insight could be gained [36]. The interview guide consisted of nine questions which were used to ensure that the research questions for the study were answered sufficiently. The guide was developed by the lead author with input and feedback given by the research team and was then validated by the full research team (paper co-authors) once the recommended changes were made. Interviews began by developing rapport between the researcher and participant, demographic information was then attained regarding the participants time working in the industry and details of their role. Participants were then asked to discuss their perceptions of current use of support services and how the support service is received. Perceptions of barriers preventing jockeys from using support services was explored by asking participants what is currently standing in the way of jockey's support service use and further enquiry was about what needs to be changed was asked. Perceived facilitators to the use of support services wase explored by asking participants their thoughts on what a support service for retiring jockeys should include. Participants were asked what would encourage this change and how could it be applied within the Irish horseracing industry. Probing questions, such as 'can you expand more on that point' and 'is there anything else you would like to add' were used at the researcher's discretion to gain further insight into topics discussed. This was followed by giving the interviewee time to add any extra information of thoughts they may have on the topic.

A pilot interview was carried out with a service provider within the industry who did not meet criteria to partake in the study as they had only recently began working for their organisation, but had keen interest in the research. Following this pilot interview, small refinements were made to wording of the script to ensure greater clarity. Interviews (n = 11) then commenced and were carried out on Zoom (Zoom Communications, Qumu Corporation, Minneapolis, Minnesota, Unites States). Zoom was considered a viable option for carrying out the interviews due to its ease of use and cost effectiveness [37]. Interviews ranged from 16–48 minutes (Mean = 29 minutes) in duration. The variance in interview duration was based on participant knowledge, or indeed lack of knowledge around what is currently available with the industry. Participants were asked to ensure a quiet space and a strong internet connection prior to the interview. Once the interviews were completed, they were transcribed and coded and the audio recordings were deleted to ensure anonymity of participants.

## Data coding and analysis

Reflexive Thematic Analysis (RTA) [34] was used in this study to analyse the qualitative data collected by semi-structured interviews with the researcher's subjectivity seen as being beneficial [34]. To begin, the lead author became familiar with the data set by reading and re-reading the interview transcripts, notes were handwritten to ensure familiarisation. Notes were taken casually at this phase and there was no specific systematic approach, some notes were words and others were short sentences. Notes were then collated from each of the individual interviews to complete the familiarisation stage to capture the overall observation that might be useful for guiding coding. Inductive methods were used in the initial stages of data coding as the codes (barriers n = 24, facilitators n = 15) developed were taken directly from the raw data itself. Deductive methods were the applied used to organise data into themes and sub themes based on how the data codes related to the research questions (barriers and facilitators to support service use). Codes were reviewed and discussed with two members of the research team to support reflexive engagement with the data and to consider alternative interpretations. Themes were then reviewed, refined, named, and organised into themes and sub themes. A thematic map for each theme was then developed to help tell the story of the themes and to ensure a visible framework. Finally, this paper was written to report on the data collected within the results and discussion sections.

## Rigor

Peer debriefing [38] was used to ensure quality standards throughout the research process. Peers (study supervisors) were given an opportunity to cast a critical eye over the project at various stages including development of the interview guide and write up of this paper. Consistent with reflexive thematic analysis, the researcher acknowledges their active role in generating meaning from the data. Familiarity with the horseracing context informed both engagement with participants and interpretation of their accounts. Reflexive practice was maintained throughout the analytic process, with ongoing consideration of how prior knowledge and assumptions shaped analytic decisions and theme development.

Given the small and interconnected nature of the Irish horseracing industry, potential power dynamics between the researcher and participants should be acknowledged. The researcher's familiarity with the sector may have influenced how comfortable service providers felt discussing limitations in current support provision or barriers affecting jockeys' engagement with services. Participants may have moderated their responses to protect organisational relationships or reputations within the industry. To reduce this influence, confidentiality and anonymity were emphasised, and participants were informed that the research was independent of governing bodies; however, this potential influence should be considered when interpreting the findings.

## Results

Two core themes were developed as defined by the research questions, reflecting on (a) perceived barrier to jockey's use of current career transition support services and (b) perceived facilitators to jockeys using career transition support service

in the future. Higher order and lower order themes are presented on thematic maps (Figs 1 and 2). Results are also presented in narrative form with direct quotes from participants included for depth and rigor.

## Theme 1: Perceived barriers to jockey's use of career transition supports services

### Higher order theme 1: Culture within the Irish horseracing industry

**Subtheme 1: Stigma towards using support services.** It was mentioned in several interviews that there is a perceived stigma present amongst jockeys to help seeking in general, and that planning an alternative career may be perceived as being uninterested and lacking commitment as a jockey. This perception is highlighted in the quote below, with the participant stating that the mindset of jockeys needs to change so that the stigma that has been imbedded can be replaced. It was clear across the interviews that the identity of jockeys is solely focused on their career as an athlete and that this mindset could be preventing them from accessing and using support services.

> "The whole mindset has to change because the jockeys of old would take the view that there can be no back door as a jockey if you want to be a top class pro and trying to get them to understand that in fact they will ride better and with more confidence and more belief in themselves if there actually is another career waiting for them and if this is a stepping stone to the next stage. That is a mentality change that is only slowly being embraced". (Participant 11).

The culture within the Irish horseracing industry is perceived to be one whereby jockeys feel that they cannot partake in anything other than their horseracing career. Negative perceptions of fellow jockeys and horse racing trainers (jockey employer) were perceived as the main reasons for this stigma. The relationship a jockey has with

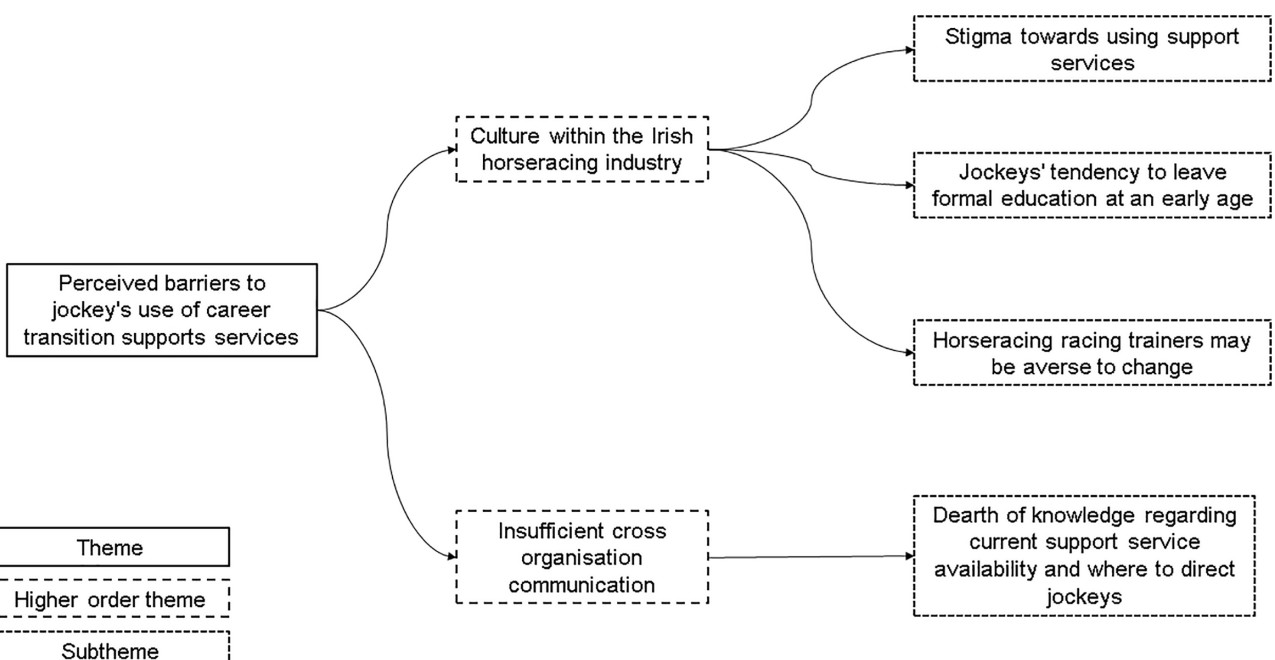

**Fig 1. Perceived barriers to jockey's use of career transition support services.** Thematic representation of perceived barriers influencing jockey's engagement with career transition support services.

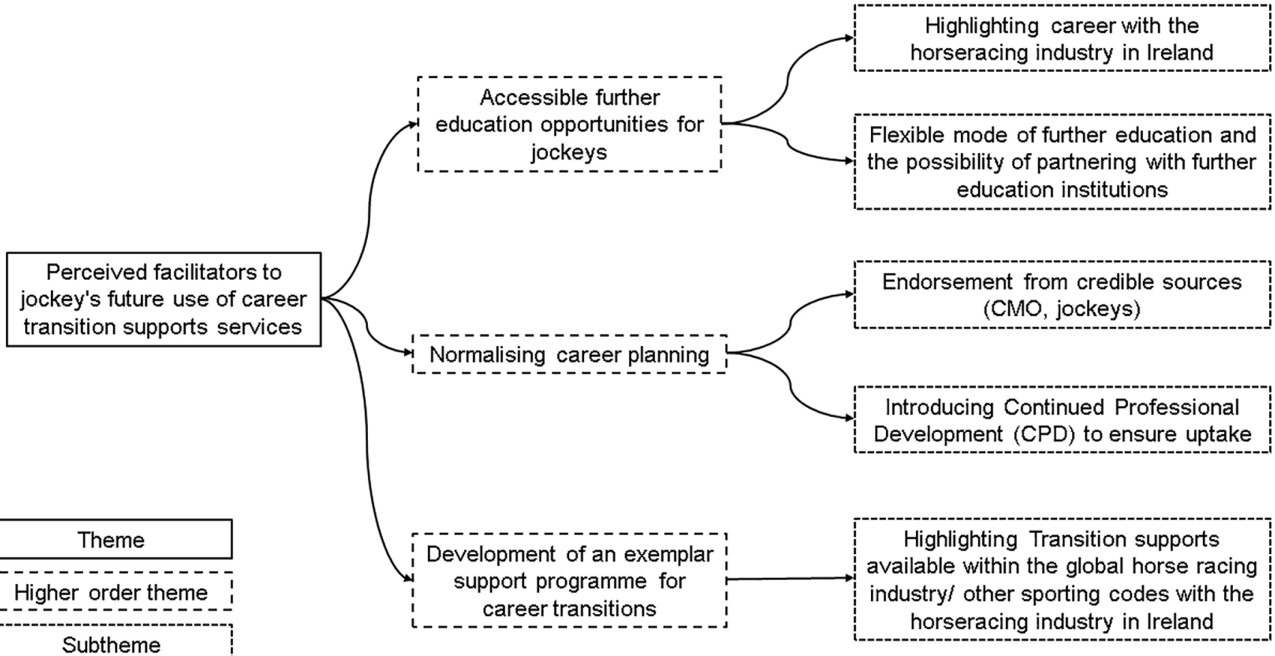

**Fig 2. Perceived facilitators to jockey's use of career transition support services.** Thematic representation of perceived facilitators influencing jockey's engagement with career transition support services.

the horse racing trainer is of vital importance to their career and employment. One participant mentioned this by stating that:

> *"When you're in the job and people have the perception of out retiring you won't get any rides. They'll say sure he's thinking of leaving, he's not in the game anymore"* (Participant 5).

**Subtheme 2: horseracing racing trainers may be averse to change.** The influence that the trainer has on the career of a jockey was evident throughout the course of the interviews, whereby a sense of fear can prevent jockeys from furthering their career or transitioning, as a jockey may lose rides and therefore their livelihood. This perceived aversion of trainers to change was mentioned on several occasions and was mentioned as being a cultural barrier that could prove detrimental in trying to support jockeys and prepare them for their retirement transition. This quote highlights the influence that trainers have on the career choices and career planning of jockeys and highlights adversity to change.

> *"One of the jockeys we're working with said they would love to set up their own business on the side while riding for just one or two trainers but the second they do that they are seen as not committed and will lose rides from that. There's one jockey that was very successful, I don't mean to stay harping on but he's gotten an opportunity to set up a racing yard and they've gotten a few really good customers and he say that he can't ride once or twice a week for people that he knows and trust because he's fearful that the second he does that he will be seen as being 'on the way out' and he won't be used by trainers. It's very hard to get the balance and continue your work career alongside your sporting career. I think any stakeholders would be very harsh on anyone who tries to do that"* (Participant 2).

The influence that the horse racing trainers have on the lifestyle decisions of jockeys was mentioned throughout the course of the interviews alongside the aversion of trainers to change the way in which their working system operates. When asked about potential barriers impeding jockeys from using support services, on participant stated:

*"Is it welcomed by everyone? I would have to say not, and I would have to say for example that trainers are quick to avail of a less educated workforce that are readily available to them, and it suits them to have these guys in the yard that are not overly ambitious"* (Participant 11).

Irish professional jockeys are dependent on the trainer to get race rides and gain employment and therefore the jockey may feel obligated to ensure the trainer is content with their life choices and career decisions. This point was made clear across a number of interviews and is highlighted by one participant stating;

*"I don't want to sound pessimistic in relation to this, but I think you will get trainers who employ the services of a jockey and in 99% of cases the trainer has very little interest and shows very little support for anything like that* (dual career) *when it come to the jockey because it's not what they care about. They only care and want this guy or girl to be there in the morning to ride horses and anything that's taking them away from that or removing them from total commitment to what they want – they don't see that as a positive – they see that as something that is negative and unfortunately, that's the way in my experience that the industry works".* (Participant 1).

**Subtheme 3: Jockeys tendency to leave formal education at an early age.** Jockeys tend to leave education at an early age. In Ireland an individual can leave formal education after completing their Junior Certificate Stat Examination. This exam falls at the end of their year of second level education, roughly around the ages of 15–16 years.

*"You need to remember the cohort you're dealing with, in many cases they have left mainstream education at a young age and in most cases – yea I think it's fair to say they wouldn't have had a positive experience of education. They were happy to be out of the classroom – they're more practically orientated. Education to them will sound like going back to school and school was a negative thing for them so that will be a barrier"* (Participant, 1).

The culture within the horse racing industry in Ireland was mentioned as a reason for jockey's lack of participation in further education and career planning. It is possible to gain employment within the horse racing industry as a jockey without completing the Leaving Certificate or any third level education. It is perceived that jockeys in general either did not enjoy the education system in Ireland or are so embedded in racing that getting a license early takes priority and therefore choose to leave formal education early due to the availability of full-time employment in the horse racing industry;

*"…you've got a situation that's unique to jockeys and racing because a lot of these guys are plucked out of education at an early stage – well they're not plucked out, they want to get out, it doesn't suit them for whatever reason and so they find themselves at 16/17 dropping out of full-time education to go into racing and that in itself creates issues."* (Participant 11).

**Higher order theme 2: Insufficient cross organisation communication**

**Subtheme: Dearth of knowledge regarding current support service availability and where to direct jockeys.** Communication issues within the Irish horse racing industry were cited as being an on-going issue. Cross organisation communication and insufficient knowledge of current support services were stated frequently throughout the interviews. One service provider stated that there are *"probably services there"* but they were unsure as to where to

go to look for them. This highlighted the lack of cross organisation communication in the area. Another service provider interviewed stated;

*"There is no proper structured career service within the racing industry"* (Participant 1).

The quote above shows the disparity that currently exists within the industry regarding current support services available. One interviewee said that the jockeys should clearly know who to contact if they need advice on careers but at the moment, they have no idea where to go;

*"In my view the industry needs a coherent hub, which involves a careers thing really. It starts engaging with people from the moment they enter the industry, and in the case of license holder, from the moment before they get their first license. And so yea, I think we don't have it currently. We have fragmented stuff that has been developed ad hoc, that looks after certain niches, but it's not coherently drawn together into a recognisable service. If you ask anybody in the industry who they would pick up the phone to if you need help on careers – that answer should be a no brainer, there should be one."* (Participant 1).

Based on these findings there is a severe dearth of knowledge about what currently exists within the horse racing industry pertaining to career transitions of professional Irish jockeys and it seems that there is no coherent support service available to aid career transition. This was also evident in this particular study as many service providers lack clarity regarding availability of support services and are therefore unable to direct help seeking jockeys;

*"No I still don't know if there's anything there to be honest"* (Participant 5).

Service providers in this research felt that the jockey and the stakeholders would not know where to find support services as there are too many different organisations offering different elements of support.

"*There are supports for them, we have supports in the Pathway and then with the Injured Jockeys Fund it's just the lack of cohesiveness and understanding of who to go to and when to go to these people and what does the retirement pathway look like?"* (Participant 9).

### Theme 2: Perceived facilitators to jockey's future use of career transition supports services
**Higher order theme 1: Accessible further education opportunities for jockeys**

**Subtheme 1: Flexible mode of further education and the possibility of partnering with further education institutions.** The implementation of a flexible model of education was discussed to allow jockeys to use their quieter periods in the racing calendar and possibly when injured to pursue further education courses.

Education was mentioned on numerous occasions in the interviews as being a facilitator to the use of support services. The options for the type of education made available and the methods of ensuring uptake were mainly discussed by service providers.

*"Have the programmes and opportunities there for them when they're in the off season so they can avail of it then or when they're injured"* (Participant 2).

Partnering with an educational institution was also mentioned as well as having regional courses to make it easier for the jockeys to partake.

Participants in this study believed that a flexible approach was the best way forward;

*"Ok so straight off that will be received negatively – this is something that we need to change with the culture. We need to adapt and get courses on board so with business for example we get someone who is able to run a course at times that suit so for examples during the afternoons, midweek evening – avoid the mornings and weekends because that's the jockeys' busiest times. If you could adapt it that way because now we've gone Zoom crazy they could be recorded sessions so if they can't make it, they can watch recorded sessions and talk to their lecturer afterwards. If we look like we are trying to work around the yards it will be received more positively"* (Participant 9).

*"There's loads of examples of sports people who can do it at the highest level"* (Participant 8).

This quote shows that that support service providers within the industry are aware that further education and dual careers are possible for jockeys. The busy work schedule of jockeys should not be overlooked however and jockey perceptions of time availability to pursue further education should be explored in future research.

**Subtheme 2: Highlighting career with the horseracing industry in Ireland.** Jockeys are passionate about their sport and therefore pursuing a career outside of the industry may be viewed as impossible or uninteresting. Throughout the course of the interviews many of the participants highlighted the need for within industry careers to be made accessible;

*"Show them jobs that are racing connected"* (Participant 9).

Within Horse Racing Ireland (HRI) a careers map has been developed which clearly shows the roles that are available within the industry with the aim that jockeys can find information about how these qualifications can be gained. The within industry careers map is currently available for jockeys however one participant also stated that: *"we need to take away the stigma for those that go back to do something else to make a good living. Look the services are there we just need to get them to the jockeys and parcel them in a different way".* (Participant 3).

**Higher order theme: Normalising career planning**

**Subtheme 1: Endorsement from credible sources (CMO, jockeys).** Interviewees mentioned the need for credible sources to endorse using support services to ensure jockey buy in. When asked about perceptions of Irish jockey's use of current support services one participant stated:

*"Not as well as it could do; Ehm, there's a couple of reasons for that, one is that its early stages so most don't know it exists. They maybe know about it when they go to someone like the sports psychology service provider who would be great at recommending. Interestingly the nature of jockeys would be that they are more likely to take a recommendation from someone they trust. But going on a website and doing some research themselves might not be how to connect with a service but if it's recommended by someone, they believe has their best interests at heart it's easier to reach out".* (Participant 7).

The influence of the Irish Horse Racing Regulatory Board (IHRB) Chief Medical Officer (CMO) and prominent Irish jockeys both current and retired was mentioned in several interviews as being an important role in the uptake and enhancement of the use of support services. The CMO is a role in which there is a lot of contact on the ground with the jockeys and was Endorsement from the CMO as well as successfully transitioned jockeys regarding career planning could potentially lead to a psychologically safe environment for jockeys. The CMO was perceived by many participants as being someone the jockey's trust. Endorsement of career planning from the CMO of the IHRB, successfully transitioned jockeys and support

service providers could ensure that jockeys feel secure and free of fear when exploring career planning opportunities in the future.

The development and availability of case studies of jockeys who have pursued further education, training and career planning may be useful in the promotion of these support services in the future;

*"If people see that jockey X has a degree but they're also a very talented rider"* (Participant 2).

*"I just think horseracing is quite traditional in terms of what do you need all that for to ride a horse. I just don't think there's a very well-rounded understanding of what education is, you know. There are people out there like Rachel Blackmore who is doing so well and also has her education behind her. And if that happen more people can start to see that oh right, she has a degree but she still a very talented rider". (Participant 6).*

One participant noted that highlighting jockeys who have pursued further education successfully could break down the barriers that currently exist. Case studies of jockeys who have successfully partaken in further education or part time employment could be beneficial to encourage others to do the same. The development of a role for successfully transitioned jockeys to act as coach-mentors to share their stories could aid in the uptake of transition support services in the future and could work alongside the CMO in the promotion of these support services.

### Subtheme 2: introducing Continued Professional Development (CPD) to ensure uptake

The introduction of CPD as part of a professional jockey's annual license was highlighted on many occasions over the course of the interviews to ensure use of support services. This would lead to a need for policy change within the industry which would ensure jockey uptake. The idea of jockeys gaining 'credits' annually as part of licensing would mean that they could potentially choose support services/ further education modules that may be applicable to them at a certain stage of their career;

*"Choose CPD that is relevant to problems they are having at that time"* (Participant 8).

The introduction of annual CPD would also encourage jockeys to engage in career planning at an earlier stage in their horse racing career which would subsequently lead to a higher level of preparation by the time they face the retirement from sport transition;

*"It's important to get in early with these guys to encourage them to invest in their future"* (Participant 2).

Within the industry there is a Jockey Pathway service available to jockeys including sports psychology, nutrition, career planning and strength and conditioning services. CPD could help to foster and expectation around further education and use of support services. The implementation of CPD could enhance the use of these support services and negate the perceived stigma that has been mentioned in the barriers section by normalising such behaviours;

### Higher order theme 3: Development of an exemplar support service for jockey career transitions

**Subtheme 1: Transition supports available within the global horse racing industry/ other sporting codes.** A number of interviewees discussed the need to look to other sports to gain insight into the transition support services they have available to their athletes;

*"It doesn't have to cost the earth do implement something very simple. I think it will be very interesting to see what you land at, at the end of all of this. I'll also be interested to see what you find in other sports – I think we need to be ambitious"* (Participant 2).

Many interviewees gave the example of the Jockeys Education and Training Scheme (JETS). This is a support service that British horseracing offer their jockeys to support them throughout their riding career and in retirement. They aim to aid with career development during their career as a jockey's and they help to prepare the jockey for life after racing.

*"There are a lot of challenge's but again I think the template they have shown in JETS is one that could be adapted but the principles could work really well in Ireland if we could get our act together"* (Participant 1).

When asked if a specific retirement support service should be implemented for Irish jockeys, one participant yet again mentioned JETS in the UK, saying:

*"Very much so. A lot of jockeys come into racing at a young age. Many have left school after national school and work in stables and yards. Very few have second level education, and it would be uncommon. I think something equivalent to JETS in England because that is a superb service. It re-educates jockeys and helps them down different pathways and is very badly needed". (Participant 4).*

## Discussion

The present research identified four higher order barrier themes (Irish horseracing culture, horseracing trainers' aversion to change, jockeys leaving formal education at an early age and insufficient cross organisation communication) and three higher order facilitator themes (accessible further education opportunities for jockeys, normalising career planning and the development of an exemplar support service for jockey's career transitions). The following section will reflect on how these findings advance previous research and the implications of these finding for practice.

Research has shown that athletes who prepare for retirement are more likely to have a better transition out of sport [39], however it has also been noted that even when support services are available the culture within a sport can create stigma and prevent athletes from buying in [1]. Culture shapes the way athletes feel, think and act [40] and therefore a person's actions and psychological processes are constituted through the cultural understandings they have been embedded in [40,41]. The perceived culture of isolation that has been mentioned as being part of the horse racing community should be explored further and potential facilitators to a change in culture should be discussed with both jockeys and stakeholders.

The culture that has developed within Irish horseracing was described by the participants as being one in which the jockeys tend not to seek help or guidance. It was mentioned in a number of interviews that there is stigma present amongst jockeys to help seeking in general, and that planning an alternative career may be perceived as being uninterested and lacking commitment. Theoretically this perceived lack of commitment and stigma can be linked to athletic identity. Athletic identity is defined as the degree to which and athlete identifies with their role as an athlete [42]. Research has found that higher levels of athletic identity can have a negative effect on career planning in athletes [22]. Within the jockey population there is a perceived stigma present towards planning for the future or having other interests outside your work as a jockey which highlights the strength of athletic identity in the jockey population. The level of athletic identity should be explored and measured in jockeys as it may be preventing them from accessing support services throughout their career.

The relationship a jockey has with the horse racing trainer is of vital importance to their career and employment. Research has found that social support and trusting relationships develop psychological safety [43,44] within the workplace and more recent research has shown its importance in sport [45,46]. This culture toward help seeking should be explored further by interviewing jockeys to establish their perspective on low levels of support service use. The culture towards help seeking needs to be addressed by support service providers and governing bodies within the industry to ensure that jockeys are psychologically safe [45] and prepared for their career after horse racing.

Horseracing trainers' aversion to change was mentioned throughout the interviews as a perceived barrier to jockey's use of support services. There was a clear perception present that horseracing trainers may not be receptive to changes being made that might encourage jockeys to use support services to help with career planning and transition planning. In the sport of horse racing, the horse trainer is the person who has a role like a coach in other sporting codes. Research has found that coaches have an important influential role on the athletes that they work with [47,48] and the consequences of this influence should not be overlooked. Research has suggested that perhaps the potential for coaches' level of power and domination may be under theorised [49]. Research with young athletes in a talent development programme has found that coaches are in a position whereby they can either inhibit or promote access to talent development environments [50]. The influence the coach has on athletes has also been documented by Arnold and Fletcher [51] who found organisational culture within sport to be a stressor for athletes and these cultural issues were deemed to be synonymous with the leadership of the coach. Irish professional jockeys are dependent on the trainer to get race rides and gain employment and therefore the jockey may feel obligated to ensure the trainer is content with their life choices and career decisions. A study by King [21] reported that trainers are an interpersonal stressor for Irish professional jockeys and therefore their aversion to change could have a knock-on effect with the jockeys themselves. This study has found a perceived lack of support from horse racing trainers regarding jockey's use of support services while King [21] also found horse racing trainers to be an interpersonal stressor for the jockey population. This link should be explored further within research and thoughts of professional jockeys themselves should be further explored.

Historically jockeys tend to leave formal education at a young age, and this is seen as a perceived barrier for the use of support services for career transition planning [21]. Research [21] has found that Irish professional jockeys (n = 84) are early school leavers with 45% leaving school after completion of the junior certificate state examination which occurs at roughly fifteen to sixteen years of age in Ireland. The same study showed that only 9% of the jockeys studied or attended third level education. Once again, the culture within the horse racing industry in Ireland was the perceived reason for jockey's lack of participation in further education and career planning. It is possible to gain employment within the horse racing industry as a jockey without completing the leaving certificate state examination or any third level education. It has been found that lower levels of education are linked to higher rates of anxiety and depressive disorders and lower rates of help seeking for these conditions [52]. Research in Finland has found that people with low school motivation may experience higher motivation in other life domains and this may create a pulling effect away from school [53]. This study linked early school leaving to self-determination theory [54] whereby early school leavers may exit the school ecosystem early to seek need satisfaction elsewhere. Irish jockeys leave formal education at an early stage [21] in comparison to the public and this could be linked to their passion for horse racing and the availability of employment in the area. A systematic review investigating the factors that influence early school leaving [55] found long working hours and attractive wages as reasons for teenagers leaving formal education early. This should be explored further with the jockeys themselves.

Pre-planning by use of further education has been suggested as means to ensure the athlete feels autonomy over their retirement transition [16,28,30,48]. It has been demonstrated in other elite level sports there is a need to move the approach from dealing with traumatic transitions to pre planning for the retirement transition [3,16,26]. This is the option taken by a vast number of jockeys. Lower levels of education due to leaving school at a young age may lead to jockey's academic self-efficacy being at a lower level to the general population. Self-efficacy is an important internal factor in smooth transitions away from sport. Academic self-efficacy is the ability of a student to master the academic subjects they are partaking in [56]. In relation to career transition this may mean an athlete being competent to perform a specific task or behaviour to gain a desired outcome. For jockeys this might be developing a curriculum vitae or preparing for interview. Further education and support services could look at ensuring these needs are met academically and that the jockeys might be more confident in pursuing a further career away from the horse racing industry. Returning to education may be necessary for jockeys to ensure that they can transition to their next career successfully and this may be challenging if they were early school leavers. It has been found that early school leavers are more likely to return to education

if it is linked with finding satisfying work in the future [57]. Engaging with support services for career planning throughout their horse racing careers to pre plan for their next career may be beneficial to aid in motivating jockeys to pursue further education.

As outlined in the methods section, there are support services available to Irish jockeys for career planning. However these support services are not specific to the retirement transition, and it was evident in the interviews that not all participants were aware of the current support service availability. Based on these findings there is a severe dearth of knowledge and lack of cross organisation communication was evident regarding what currently exists within the horse racing industry pertaining to career transitions of professional Irish jockeys. This lack of knowledge about support services and poor cross organisation communication needs to be addressed in line with the recommendations outlined in both the 2009 and 2021 ISSP position stand on career development and transitions of athletes [32,58]. These documents state that sporting organisations are expected to provide more support to athletes during their retirement from sport [32,59]. Athletes in the past have lacked awareness of services outside of their own organisation [60] which could lead to loss of use of career transitioning services. Knowledge on career services within the industry are of vital importance going forward to ensure use of these services. Highlighting the current support services available to career transitioning jockeys should be the first step to ensure the jockeys are aware that help is available to them.

Service providers in this research felt that the jockey and potentially the service providers themselves may not know where to find support services as there are too many different organisations offering different elements of support. Previous research regarding mental health help seeking states that for issues to be addressed the initial step is knowing where to find help [61]. Similarly, studies have found that lack of a clear route that is easily accessible for athletes is a barrier to the use of support services [62,63]. However, there appears to be no direct route for support within the horse racing community. This is similar to a study of elite athletes which found that they lack awareness of where to find planning services outside of their own organisation. This highlights the possibility that if the governing body for horse racing in Ireland does not provide coherent and easily accessible transition services, a barrier to the use of support services could exist. Research has shown [21,62,64] that ease of access and visibility of services is a key determinant to use of support services for help seeking jockeys in relation to mental health services and it is perceived that a similar problem is evident for career transitioning jockeys. This issue could be alleviated by exploring the jockey's perceptions on the forms of communication that work best them.

The implementation of a flexible model of education was discussed to allow jockeys to use their quieter periods in the racing calendar and possibly when injured to pursue further education courses was mentioned as being a facilitator for use of support services. This could alleviate the problem of lacking formal qualifications when the retirement transition occurs which has been found to be troublesome in other sports [21]. Educational status upon retirement from sport has been found to lead to occupational difficulties when seeking employment, whereas athletes who had a higher level of education experienced less occupational difficulties in a study of Slovene international and national level athletes [17].

Partnering with an educational institution was also mentioned as well as having regional courses to make it easier for the jockeys to partake. This was also recommended in a European review of university education for elite athletes [65] which stated that that a requirement should be instated via legislation in academic institution to allow elite athletes to have adapted. These flexible approaches might help ensure uptake of available support services by jockeys who have a busy year-round calendar of racing with long and irregular working hours and variable individual schedules [66,67]. Such options could alleviate the issue of time constraints which past research has stated athletes find the most difficult when pursuing further education or a dual career [68]. de Subijana et al. [68] also stated the need for a sports psychologist to work with elite athletes that are returning to education to teach them the time management skills necessary to aid success both athletically and academically. Facilitating athletes to be able to partake in further education and training during their horse racing career is vital to aid in the retirement transition [24]. Jockeys need to be shown that two careers are mutually complementary and that in fact it is encouraged by EU directive as best practice [65]. Previous research has suggested

that athletes are better to have multiple identities including social, academic, and occupational interests outside of their sport [28,59] to ensure overall wellbeing. The perception of the jockeys themselves needs to be gained in future research to investigate their thought on this subject matter [28,59,67,69–71].

The retirement transition can be extremely difficult to navigate for many athletes' due to a loss of athletic identity [72]. Jockeys are passionate about their sport and therefore pursuing a career outside of the industry may be viewed as impossible to some with research finding that life satisfaction and athletic identity can be negatively affected leading up to and after the retirement transition [16]. Therefore, it was suggested in the interviews that highlighting the careers that are available to jockeys within the horseracing industry itself could be a good starting point in encouraging them to partake in career planning during their career. A visible career coach/ well-being officer on the ground who can easily direct jockeys to the correct support services should be explored to alleviate these issues. Career support and lifestyle management support have been suggested to enable elite athletes to balance sport, education and vocational and personal lived effectively to ensure well-being [73]. These supports could play a vital role throughout the sporting career of Irish jockeys. Research has shown that social persuasion can increase self-efficacy if carried out effectively [74,75]. Normalising discussion around future career transitions and highlighting options both racing related and in other industries could help to alleviate the uncertainty that is currently faced by athletes, to allow for a smoother end of athletic career transition.

Normalising career transition planning was mentioned in many interviews as a perceived facilitator to the use of support services. The influential role of both the Chief Medical Officer (CMO) and fellow jockeys could have on the perceptions of jockeys to using career transition support services was highlighted. The influence of the Irish Horse Racing Regulatory Board (IHRB) CMO and prominent Irish jockeys both current and retired was mentioned in several interviews as being an important role in the uptake and enhancement of the use of support services. The CMO is a role in which there is a lot of contact on the ground with the jockeys and was perceived by many participants as being someone the jockeys trust. The influence that a knowledgeable and credible source [69] such as the CMO can have on the jockey's self-efficacy can be linked to Schlossberg's Career Transition Theory [33], whereby institutional support availability is a cornerstone. Research has also found positive effects of verbal persuasion from a significant person [33,76] whereby jockeys can potentially be convinced and encouraged to partake in career planning for their retirement transition. The role of CMO in Irish horse racing can be linked to the role of a team captain in other sports which has been found to play an important role in the promotion of positive life skills development among teammates [77]. This relationship between CMO and the jockeys should be explored further to maximise the potential that can be gained through this method. It should be noted however that CMO is a full-time occupation and that perhaps the development of a role to specifically work with career transitions could be developed in the future to alleviate the pressure on the CMO. Involvement in education and career planning have been found to be positively associated with post sport life adjustment [3] and therefore a role to facilitate this should be explored in the future.

This study suggested role models should be incorporated into career counselling with athletes to support them (Ronkainen et al., 2019). One participant noted that highlighting jockeys who have pursued further education successfully could break down the barriers that currently exist. The development and availability of case studies of jockeys who have pursued further education, training and career planning may be useful in the promotion of these support services in the future. A study of young (aged 17–18) Finnish athletes found that most chose role models from their sport who had other identities and who participated in other areas of life outside of the sport itself [78]. The development of a role for successfully transitioned jockeys to act as coach-mentors to share their stories could aid in the uptake of transition support services in the future and could work alongside the CMO in the promotion of these support services.

Endorsement from the professional working as CMO, as well as successfully transitioned jockeys regarding career transition planning could potentially lead to a psychologically safe environment for jockeys. Psychological safety in sport is defined as the perception that one is protected from, or unlikely to be at risk of, psychological harm [45]. Fransen et al. [79] found that leadership behaviours can strengthen athletes' identification with their teams which in turn can cultivate

psychological safety. Similarly in work settings, psychological safety has been found to satisfy the needs of workers and predict thriving in the workplace [80]. The first and most important step in creating psychological safety is for the leader to develop a positive environment [81]. Developing psychological safety within the horse racing community in Ireland would ensure that jockeys basic psychological needs of competence and autonomy [54] were met, again linking to Sclossberg's Career Transition Model whereby the situation, self, support and strategies interact to influence coping and adaption [33]. Endorsement of career planning from the CMO of the IHRB, successfully transitioned jockeys and support service providers could ensure that jockeys feel secure and free of fear when exploring career planning opportunities in the future.

The idea of jockeys partaking in CPD annually as part of licensing would mean that they could potentially choose support services/ further education modules that may be applicable to them at a certain stage of their career. The benefits of CPD are promoted by the European Union since the Lisbon Agreement (2000) which states the necessity of lifelong learning, skill development and knowledge development for all workers. Ample opportunities must be in place in an organisation to ensure a climate that will develop change and encourage individual development for learning [82–84]. This would lead to a need for policy change within the industry which would ensure jockey uptake. A culture that supports workers continued professional development has been found to be positively associated with career potential in the general workforce [85].

An applied example of CPD in sport is evident within the Canadian Sport who have developed a career transition programme which involves athletes partaking in nine online modules during their career to help them understand and prepare for the retirement from sport transition, (Introducing Game Plan Canada's Athlete Career Transition Programme, 2015). This model should be explored so that a career transition service can be developed and introduced to Irish jockeys. The introduction of annual CPD would also encourage jockeys to engage in career planning at an earlier stage in their horse racing career which would subsequently lead to a higher level of preparation by the time they face the retirement from sport transition. Within the Irish horseracing industry there is a Jockey Pathway service available to jockeys including sports psychology, nutrition, career planning and strength and conditioning services. CPD could help to foster and expectation around further education and use of support services which research has suggested can change a culture within sport [21]. The implementation of CPD could enhance the use of these support services and negate the perceived stigma that has been mentioned in the barriers section by normalising such behaviours.

Exploring transition supports available within the global horse racing industry and other sporting codes was discussed in the interviews as being a potential facilitator to the development of an exemplar transition support service for Irish jockeys. Participants mentioned the possibility of using the UK Jockey Education and Training Scheme (JETS) programme as a template of best practice, which could be adapted and adopted within Ireland, with professional jockeys. JETS is a charity organisation set up in the UK to provide supports to jockeys during their time as a jockey, and to help them to prepare for life after racing. A personal development plan is completed at the beginning of the jockey's career in racing, and this is then developed throughout the jockey's career to ensure that their potential is reached and that they are prepared to transition to their next career successfully. JETS are also linked with many employers and can direct jockeys to fill vacant positions where they can use their identified transferrable skills successfully. In 2012 European Union guidelines noted that dual careers in elite athletes should be encouraged and they stated that changes to policy including education, health, employment, and finance should be mobilised [40]. Further research should explore the jockey's perceptions on dual careers as the workload involved in the sport is intense and therefore this might not be a possibility. It has also been suggested that athletes should look to their governing body for guidance on career transition [86] which highlights the need for these governing bodies to have a career transition support service in place. An example within Ireland is the work of Rugby Players Ireland (RPI) who provide career planning, further education, and personal skills development to help professional rugby players be successful on and off the field, (Rugby Players Ireland, 2021). These governing bodies have a visible and well-known service available to transitioning athletes which eliminates confusion and uncertainty for these athletes when they are planning for retirement. The clear and identifiable nature of these support services make it accessible for athletes to make use of their provisions.

It is evident that many successful supports service programmes already exist within the sports industry in Ireland and internationally. These support services could be used as a working template which would alleviate some of the work involved in the design and implementation for the Irish horse racing industry. Sport's governing bodies need to be made aware of best practice as outlined by an ISSP Position Stand on career development and transitions of athletes [32] to ensure transition supports are available for athletes [19]. The findings of this study are specific to Irish horseracing; however, they may be transferable to similar high-performance sport environments characterised by strong cultural norms, hierarchical relationships, and early specialisation. It is still advisable that differences in organisational structures and cultural contexts across jurisdictions and sports should be considered.

## Study limitations

Several limitations should be considered when interpreting the findings of this research. First, this study reflects the perspectives of service providers and does not include the voices of trainers, current and retired jockeys. While participants offered insight into the availability of support services, their perspectives may not fully capture how these services are perceived or experienced by the jockeys themselves. Future research incorporating jockey perspectives would provide a more comprehensive understanding of service accessibility and engagement. Second, because participants operate within the system responsible for delivering these supports, there is potential for insider bias that may shape how services and potential barriers, and facilitators are described. Third, the findings are situated within the specific organisational and cultural context of Irish horseracing, which may limit transferability to other racing jurisdictions or sporting contexts. Finally, while coding procedures were used to enhance analytic rigour, qualitative interpretation remains subject to potential confirmation bias.

## Conclusion

Preplanning for the retirement transition from sport has been found to lead to a smoother and more successful transition. This research develops this idea within the context of Irish horseracing jockeys. A of compilation of currently available support services was gathered by interviews with all support services providers from all known stakeholders within Irish horseracing. This research has also found the perceived barriers and facilitators to the use of these support services by Irish jockeys. To the authors knowledge this study is the first of its kind to explore industry perceptions of barriers and facilitators to the use of support services by Irish Jockeys in relation to retirement from horse racing. The culture within the sport of Irish horseracing and insufficient cross organisation communication were the main perceived barriers by industry service providers identified to the use of support services. It is important for an individual working within the horse racing industry and those providing support services to be made aware of these barriers so that changes can be made to make the Irish jockey more likely to use these support services going forward. The perceived facilitators in this study were accessible further education opportunities for jockeys, normalising career planning, and the development of an exemplar support programme for career planning. These should be explored further by gaining insight into the jockeys' views on this topic. The research team have committed to carrying out this qualitative research and have the support of the IHRB to do so. Both current and retired jockeys will be interviewed to explore their perspectives. Exemplar programmes to aid in career development and transitions used in other sports should be explored further to gain an in depth understanding of what they provide. This could then be used to guide the implementation of purpose specific multi-disciplinary support service for professional Irish jockeys in the future. The involvement of the IHRB in this research highlights the willingness on their behalf to implement recommendations made. Further research is needed to explore potential barriers that may arise on use of support services based on both jockey and trainer perspectives.

## Supporting information

**S1 Appendix File interview guide.**
(DOCX)

## Author contributions

**Conceptualization:** Laura Langton, Sarah Jane Cullen, Giles Warrington, Jennifer Pugh, Adrian McGoldrick, Ciara Losty.

**Data curation:** Laura Langton.

**Formal analysis:** Laura Langton.

**Investigation:** Laura Langton.

**Methodology:** Laura Langton.

**Project administration:** Laura Langton.

**Resources:** Laura Langton.

**Software:** Laura Langton.

**Supervision:** Sarah Jane Cullen, Giles Warrington, Adrian McGoldrick, Ciara Losty.

**Validation:** Laura Langton.

**Visualization:** Laura Langton.

**Writing – original draft:** Laura Langton.

**Writing – review & editing:** Laura Langton, Sarah Jane Cullen, Giles Warrington, Jennifer Pugh, Adrian McGoldrick, Ciara Losty.

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
