## [Decision Letter · Decision Letter 0]

6 Feb 2026

PONE-D-25-48875
An industry perspective in Irish horseracing on the availability of retirement specific support services for professional jockeys and perceived barriers and facilitators to their use.
PLOS One

Dear Dr. Langton,

Thank you for submitting your manuscript to PLOS ONE. After careful consideration, we feel that it has merit but does not fully meet PLOS ONE’s publication criteria as it currently stands. Therefore, we invite you to submit a revised version of the manuscript that addresses the points raised during the review process.

We look forward to receiving your revised manuscript.

Kind regards,

Hesam Ramezanzade, Ph.D

Academic Editor

PLOS One

Journal Requirements:

3. We note that you have referenced “Lynch, 2006” which has currently not yet been accepted for publication. Please remove this from your References and amend this to state in the body of your manuscript: as detailed online in our guide for authors

Reviewers' comments:

Reviewer's Responses to Questions

**Comments to the Author**

1. Is the manuscript technically sound, and do the data support the conclusions?

Reviewer #1: Partly

Reviewer #2: Partly

2. Has the statistical analysis been performed appropriately and rigorously?

Reviewer #1: Yes

Reviewer #2: Yes

3. Have the authors made all data underlying the findings in their manuscript fully available?

Reviewer #1: No

Reviewer #2: Yes

4. Is the manuscript presented in an intelligible fashion and written in standard English?

Reviewer #1: Yes

Reviewer #2: Yes

5. Review Comments to the Author

Reviewer #1: Please consider revising the title so it clearly reflects that the study is written from a service-provider perspective. It would also strengthen the paper if the limitations section were expanded to openly acknowledge the absence of jockey voices, the risk of insider bias, the limited generalisability of the findings, and the potential for confirmation bias.

To avoid the study feeling incomplete, it would be helpful to explicitly commit to Phase 2, outlining a clear plan—including timeline, funding considerations, and methodological approach—for interviewing jockeys. The discussion section could also be tightened by about 30%, removing repetition and trimming content that does not directly contribute to retirement support (including the mental-health section if it is not fully integrated).

Adding a reflexivity statement would meaningfully enhance the transparency of the work by explaining how your insider role shaped data collection and interpretation. The practical recommendations would benefit from deeper detail as well, especially regarding funding, implementation barriers, strategies for engaging trainers, and realistic rollout timelines.

As minor suggestions, you may want to clarify your theoretical framework by using one or two consistent lenses (or explicitly noting the exploratory nature of the study), include a brief comparison table outlining JETS (UK), Game Plan (Canada), and the proposed Irish model, consider member-checking with participants, and reflect on power dynamics—specifically how your organisational relationships may have influenced what participants felt comfortable sharing.

Reviewer #2: Dear Authors,

Thank you for submitting your manuscript on career transition support for Irish jockeys. I enjoyed reading about this important topic and can see the value your research brings to the horseracing community.

OVERALL IMPRESSION

You've tackled a really important issue that hasn't been studied much before. Your interviews with stakeholders across Irish racing organizations provide valuable insights, and I can see this work making a real difference for jockeys facing retirement. However, there are several areas that need attention.

WHAT WORKS WELL

Your study has real strengths:

• You're the first to look specifically at Irish jockey retirement support - this fills a genuine gap

• You spoke with representatives from all the key organizations (IHRB, HRI, RACE, etc.) which gives a comprehensive view

• The barriers and facilitators you identified are clearly presented and practical

• Your recommendations (like CPD requirements and flexible education) could actually be implemented

• The quotes from participants really bring the findings to life

MAIN ISSUES TO ADDRESS

I've organized these from most critical to less urgent:

Critical Issues (must be fixed):

1. Title typo: There's a "2" in "support 2 services" that needs removing. Also, the title is quite long - try to get it under 15 words.

2. No limitations section: This is a significant omission. Every research paper needs to discuss its limitations. Please add a paragraph addressing things like: only talking to stakeholders (not jockeys themselves), the small sample size, potential bias from being insiders in jockey research, and that findings come from just one country.

3. Abstract: Your abstract is vague about what you actually found. Instead of saying "themes were identified," tell us what those themes were and give us some numbers.

4. Research questions not explicit: Right now you say you want to "explore perceptions" which is quite general. At the end of your introduction, add clear numbered research questions like:

o RQ1: What support services currently exist for Irish jockeys?

o RQ2: What prevents jockeys from using these services?

o RQ3: What would help jockeys use support services more?

5. Missing theoretical framework: You mention several theories throughout but never clearly state which one guides your study. Pick one model (Schlossberg's transition model would work well) and add a paragraph in your introduction explaining how it frames your research.

6. Methods section needs more detail:

o Create a table showing your 11 participants (their roles, how long they've been in position, whether they work directly with jockeys)

o Explain your coding process more clearly - who did it? Was it just you or did others help? Did you check reliability?

o Your "rigor" section is only 4 lines - expand this to explain how you ensured quality (credibility, transferability, etc.)

Important Issues:

7. Organizational descriptions in wrong place: Lines 49-68 where you describe IHRB, HRI, Equipp, etc. - this belongs in your Methods section, not the Introduction. The Introduction should be about concepts and theory, not organizational structure.

8. Discussion structure: Your discussion jumps around a bit and repeats things. Reorganize it with clear sections: summary of findings, what theories explain your results, comparison to other research, practical recommendations, limitations, future research, conclusion.

9. Weak jockey literature base: You only cite two studies on jockey retirement (both Australian, both from 2006-2007). That's okay - this literature simply doesn't exist yet! But acknowledge this explicitly and explain why it makes your study even more important as foundational work.

10. Recommendations too vague: Instead of saying "stigma should be addressed," be specific. For example: "IHRB should require 10 hours of CPD annually for license renewal, including modules on career planning, financial literacy, and post-racing career options. Implementation: pilot program 2026, full rollout 2027.

11. A few typos and grammatical errors throughout (line 38: "themselves'", line 69: missing apostrophe, etc.) - needs a careful proofread

12. The philosophical section (lines 123-140) is quite long and uses jargon without defining it. Shorten to 4-5 sentences and explain what "emic epistemology" actually means for your study.

13. Discussion could be shortened , there's some repetition that could be cut.

6. PLOS authors have the option to publish the peer review history of their article (what does this mean?). If published, this will include your full peer review and any attached files.

Reviewer #1: No

Reviewer #2: No

---

## [Author Response · Author response to Decision Letter 1]

25 Mar 2026

Dear Editor,

Thank you for the opportunity to revise and resubmit our manuscript. We are grateful to you and the reviewers for your constructive and insightful feedback, which has helped to significantly strengthen the quality and clarity of the paper.

We have carefully addressed all comments raised by the reviewers and have revised the manuscript accordingly. A detailed, point-by-point response outlining how each comment has been addressed is provided as a document in the attachments section. Revisions have also been clearly indicated within the manuscript.

We believe these revisions have improved the rigour, clarity, and overall contribution of the study, and we hope that the manuscript is now suitable for publication in your journal.

Thank you for your time and consideration. We look forward to your feedback.

Yours sincerely,

Laura Langton

---

## [Editor Report · Decision Letter 1]

12 Apr 2026

A service provider perspective in Irish horseracing on the availability of retirement specific support services for professional jockeys and perceived barriers and facilitators to their use

PONE-D-25-48875R1

Dear Dr. Langton,

We’re pleased to inform you that your manuscript has been judged scientifically suitable for publication and will be formally accepted for publication once it meets all outstanding technical requirements.

Kind regards,

Hesam Ramezanzade, Ph.D

Academic Editor

PLOS One

---

## [Editor Report · Acceptance letter]

PONE-D-25-48875R1

PLOS One

Dear Dr. Langton,

I'm pleased to inform you that your manuscript has been deemed suitable for publication in PLOS One. Congratulations! Your manuscript is now being handed over to our production team.

Kind regards,

on behalf of

Dr. Hesam Ramezanzade

Academic Editor

PLOS One